# Analysis of Metabolic Components of JUNCAO Wine Based on GC-QTOF-MS

**DOI:** 10.3390/foods12112254

**Published:** 2023-06-03

**Authors:** Jinlin Fan, Zheng Xiao, Wanwei Qiu, Chao Zhao, Chao Yi, Dongmei Lin, Zhanxi Lin

**Affiliations:** 1National Engineering Research Center of Juncao, Fuzhou 350002, China; m15880180003@163.com (J.F.); lindm_juncao@163.com (D.L.); 2College of Life Science, Fujian Agriculture and Forestry University, Fuzhou 350002, China; 3Institute of Agricultural Engineering and Technology, Fujian Academy of Agricultural Sciences, Fuzhou 350013, China; 4School of Life and Health Science, Anhui Science and Technology University, Chuzhou 233100, China; 5College of Marine Sciences, Fujian Agriculture and Forestry University, Fuzhou 350002, China

**Keywords:** JUNCAO wine, metabolic components, GC-QTOF-MS, metabolomics

## Abstract

JUNCAO wine fermentation metabolites are closely related to the final quality of the product. Currently, there are no studies of dynamic metabolite changes during fermentation of JUNCAO wine. Here, we used gas chromatography quadrupole time-of-flight mass spectrometry (GC-QTOF-MS) metabolomics and multivariate statistical analysis to explore the relationship between metabolites and fermentation time. A total of 189 metabolites were annotated throughout the fermentation process. The principal component analysis (PCA) revealed a clear separation between the samples in the early and late stages of fermentation. A total of 60 metabolites were annotated as differential during the fermentation (variable importance in the projection, VIP > 1, and *p* < 0.05), including 21 organic acids, 10 amino acids, 15 sugars and sugar alcohols, and 14 other metabolites. Pathway analysis showed that the most commonly influenced pathways (impact value > 0.1 and *p* < 0.05) were tricarboxylic acid cycle, alanine, aspartic acid and glutamic acid metabolism, pyrimidine metabolism, and other 10 metabolic pathways. Moreover, integrated metabolic pathways are generated to understand the conversion and accumulation of differential metabolites. Overall, these results provide a comprehensive overview of metabolite changes during fermentation of JUNCAO wine.

## 1. Introduction

With the continuous development of the global economy, the problem of food shortage is becoming more and more serious [1,2,3]. Traditional brewing of alcohol is known to be costly and potentially a threat to food security as it mainly uses grain as a raw material. Therefore, to find effective ways to save grain, we should develop a new type of wine that can better solve the problem of competing with the people for land and grain. *C. fungigraminus* belongs to the grass family and is the most important representative of JUNCAO [4]. It has the characteristics of high yield, high fiber content, high crude protein, and high sugar content [5]. *C. citratus* is rich in polysaccharides, polyphenols, flavonoids, and citronella essential oil, and has the effect of reducing fat, lowering sugar, and inhibiting bacteria [6]. It has been found that *C. fungigraminus* and *C. citratus* contain abundant reducing sugars, providing a theoretical basis for the use of *C. fungigraminus* and *C. citratus* as a material for ethanol preparation. The finished JUNCAO wine has a strong and unique JUNCAO aroma and has a good promotion value [7].

In recent years, metabolomics has been used more and more extensively in various fields. It studies low-molecular-mass metabolites such as organic acids, fatty acids, amino acids, and sugars in biological samples [8,9]. Through high-throughput detection and data processing, information integration and biomarker identification are carried out [10,11]. Due to the importance of microorganisms in biological systems, the study of metabolomics has also received attention in the field of microbiology and has been widely used in different research fields, such as the study of microbial functional genes, microbial product metabolism, and fermentation regulation [12]. At present, the main research and analysis methods used in metabolomics are ultra-high-resolution mass spectrometry (UHRMS) [13], gas chromatography (GC) [14,15], ultra-high-performance liquid chromatography coupled with time-of-flight high-resolution tandem mass spectrometry (UHPLC/Q-TOF-MS/MS) [16], ion migration spectrometry (IMS) [17], gas chromatography time-of-flight mass spectrometry (GC-TOF-MS) [18], gas chromatography quadrupole flight time mass spectrometry (GC-QTOF-MS) [19], infrared spectrometry (IR) [20], headspace solid-phase microextraction gas chromatography mass spectrometry (HS-SPME-GC-MS) [21,22], nuclear magnetic resonance (NMR) [23], etc. Among them, headspace solid-phase microextraction and GC are widely used due to their simple operation. However, time-of-flight mass spectrometry plays an important role in the combination of chromatography with its advantages of fast detection speed, wide mass range, high ion transfer rate, high resolution, and high sensitivity [24,25]. Chen et al. [26] through HS-SPME-GC-MS and GC-TOF-MS reveal the relationship between core microorganisms and flavor substances in the fermentation process of corn wine. Similar reports in makgeolli (a traditional rice wine in Korea) fermented by koji inoculated with *Saccharomycopsis fibuligera* or *Aspergillus oryzae* found that the formations of volatile and nonvolatile metabolites in makgeolli can be significantly affected by microbial strains with different enzyme activities in koji [27].

Metabolomics analysis based on GC-QTOF-MS in biological systems and fermented foods has been shown to be effective and efficient for more comprehensive detection of volatile and non-volatile flavor substances in fermented foods [28]. The GC-QTOF-MS method is therefore potentially the most appropriate choice for primary analysis of JUNCAO wine, given the completeness and reliability of this approach. At present, there are no studies on the metabolomics of JUNCAO wine, and the application of GC-QTOF-MS is still being explored. The results help to understand the flavor formation mechanism of JUNCAO wine and provide a scientific basis to guide the production of JUNCAO wine.

## 2. Materials and Methods

### 2.1. Chemicals

Methanol, chloroform, pyridine, ammonium methoxy salt, ribose alcohol, bis(trimethylsilyl)trifluoroacetamide (BSTFA) and trimethylchlorosilane (TMCS; BSTFA/TMCS: 99/1), saturated fatty acid methyl ester, and pure water were purchased from Roe Scientific Incorporate (Newark, NJ, USA). All of these chemicals and standards were HPLC grade.

### 2.2. Wine Samples

*C. fungigraminus* and *C. citratus* were provided by Qishan campus of the National Engineering Research Center of JUNCAO. The wine fermentation agent was obtained from Angel Yeast Co., Ltd. (Yichang, China) and stored in the laboratory of the National Engineering Research Center of JUNCAO.

Fresh *C. fungigraminus* and *C. citratus* were clean cut. The ingredients were mixed in a certain proportion and then put into sterilized glass jars for inoculation and fermentation. Then, the fermented grains were separated after fermentation, filtration, and clarification. Finally, after several distillations and concoctions, the finished product was obtained. The best fermentation initial sugar content was 24° Brix and 0.45 g/L of the distiller’s yeast inoculation amount. The fermented time is 30 days, and the temperature is required to be 29 °C [7].

Fermented grains collected during the brewing time (0 days, 2 days, 10 days, 21 days, and 30 days) were put into a sterile tube, sealed, and stored at −80 °C until further analysis and marked as D2, D10, D21, and D30.

### 2.3. Untargeted Metabolomics Analysis Based on GC-QTOF-MS

The fermented grains were shipped to the Beijing Allwegene Technology Company, Beijing, China, for analysis. The sample weighed about 50 ± 1 mg and was transferred into a 2 mL eppendorf (EP) tube and extracted with 450 μL of extraction liquid (V_Methanol_:V_H_2_O_ = 3:1). A total of 10 μL of adonitol (0.5 mg/mL stock in dH_2_O) was added as internal standard, followed by vortex mixing for 30 s. The mixture was homogenized in a ball mill for 4 min at 45 Hz, then ultrasound treated for 5 min (incubated in ice water), and centrifuged for 15 min at 12,000 rpm, 4 °C. A total of 200 μL of supernatant was transferred into a fresh 1.5 mL EP tube and dried completely in a vacuum concentrator without heating. The samples treated above were mixed with 100 μL of methoxy amination hydrochloride (20 mg/mL in pyridine) and incubated for 30 min at 80 °C. A total of 100 μL of the bis(trimethylsilyl)trifluoroacetamide (BSTFA) and trimethylchlorosilane (TMCS) (BSTFA/TMCS: 99/1, *v*/*v*) was add to the sample aliquots, and the mixture was incubated for 1.5 h at 70 °C. All samples were analyzed by GC-QTOF-MS, and then the machine test was conducted in random order.

Good reproducibility is the basis of reliable data. In this experiment, the same volume of each prepared sample was mixed into a large sample and then divided into 4 QC samples for monitoring instrument precision and stability. Before sampling, 3 QC samples were used to monitor instrument precision, and one QC sample was collected at an interval of 8 samples for monitoring instrument stability.

The detection equipment used was Agilent 7890 gas chromatogram time-of-flight mass spectrometer. Agilent DB-5MS (30 m × 250 μm × 0.25 μm) column was available [29]. Gas chromatographic (GC) parameters were as follows: splitless mode, carrier gas helium, a gasket purge flow rate of 3 mL/min, a forward sample outlet temperature of 280 °C, a transmission line temperature of 280 °C, a shunt ratio of 10:1, a column flow rate 1 mL/min, and a sample injection volume of 1.0 µL. The heating process of gas chromatography was as follows: the initial temperature was 50 °C, and the temperature was kept for 1 min, then the temperature was raised to 310 °C at 10 °C/min and maintained for 8 min. The injection, transfer line, and ion source temperatures were 280, 280, and 250 °C, respectively. The energy was −70 eV in electron impact mode. The mass spectrometry data were acquired in full-scan mode with the *m*/*z* range of 50–500 at a rate of 20 spectra per second after a solvent delay of 6.17 min. 

### 2.4. Data Processing and Analysis

The GC-QTOF-MS analysis was performed using an Agilent 7890 gas chromatograph system coupled to a Pegasus HT time-of-flight mass spectrometer. The original data obtained from the GC-QTOF-MS test was preliminarily actively processed by the chromate of software (V 4.3x, LECO). The peak extraction, baseline correction, peak integration, peak alignment, and other statistical analysis were performed [30]. The Golm Metabolome Database (GMD), a metabolomics database containing a large amount of plant metabolites, was used for the comparison and screening of high-quality analytical data [31]. The volatile compounds were identified using comparison with mass spectra in the National Institute of Standards and Technology library (NIST). The ratio of peak area of target compounds to internal standard was analyzed in Microsoft Excel to obtain the relative content of metabolites. The homogenized data were imported into the LECO-Fiehn Rtx5 database for principal component analysis (PCA), including mass spectrum matching and retention time index matching, to evaluate the overall metabolome variability. The orthogonal partial least square discriminant analysis (OPLS-DA) method was used to find the different main components, and S-plot was used to compare and analyze the metabolites of fermented grains in different fermentation processes. The discriminant method of differential metabolites was the OPLS-DA model, the first principal component variable effect value (VIP) was set to be greater than 1, and the p-values of both Student’s *t*-tests were less than 0.05, which was considered a significant difference in metabolites. Finally, MetaboAnalyst was used for pathway analysis.

## 3. Results

### 3.1. Sample Quality Control and Overall Analysis of Metabolites

In the total ion flow chromatogram of JUNCAO wine flavor substances (Figure 1), the instrument’s operation is stable, and the retention time reproducibility is good. A total of 790 peaks were monitored in this experiment, and no drift was observed in all of them, indicating a stable retention time. In addition, the general trend of the total ion flow chromatogram for JUNCAO wines at different fermentation periods was approximately the same. However, the relative content of some substances was different, leading to apparent differences in peak heights. The ions missing more than 50% in QC samples or more than 80% in actual samples were removed, and then the ions with a relative standard deviation > 30% of all QC samples were filtered out. Finally, the total number of ions was 135, accounting for 54.22%. A total of 189 metabolites were detected in these samples (Appendix A).

### 3.2. PCA Analysis

Multivariate statistical methods are mainly used to distinguish and classify the measured variables through chromatography to understand patterns and identify molecular markers [32]. Regarding the results of PCA analysis in Figure 2a, it is shown that there were significant metabolic differences in JUNCAO wine at different stages of fermentation. The unsupervised PCA model showed that the contribution rate of the two principal components was 80.6% (PC1, 47.6% and PC2, 33%). The agglomeration at days 0 and 2 was the best, indicating that the metabolites of the three duplicate samples differed the least from day 0 to day 2 of fermentation, and that metabolism was not very rapid during the first two days of fermentation. Samples of fermented grains at day 0 and day 30 showed a clear separation trend, indicating that the polar metabolites are significantly different between the two groups. After the second day of fermentation, once the yeast reached a stable growth stage, yeast metabolism was vigorous, so the content of flavor substances began to rise sharply [33]. The gradual change in metabolites from 10 days to 21 days in the intermediate stage of fermentation and from 21 days to 30 days of fermentation at the end of fermentation was consistent with the gradual change of metabolites during fermentation. Samples at day 10 and day 30 were grouped together because the fermentation environment during saccharification was similar at day 10 and day 30. Part of the microbial community was active at intermediate stages. During fermentation, it decayed or died, and the microbial community returned to a microbial state at the end of fermentation on day 30. Jiang [34] performed a PCA analysis of the metabolites in fermented grains of black glutinous rice wine which showed a clear separation between the samples after 4 days and 2 days, and the samples after 4–24 days clustered together, which was consistent with the results of this study.

To find information about metabolites that cause significant differences, we further adopted a supervisory classification discrimination model, the partial least squares analysis method (OPLS-DA). The results are shown in Figure 2b. The differences between the groups can be seen in the abscissa direction (42.7%), and the differences in samples within the group can be seen on the vertical axis (10.3%). As can be seen from the OPLS-DA score plots, sample agglomeration and dispersion, on the one hand, indicated sample reproducibility but, on the other hand, also reflected the similarity of metabolic profiles among sample groups. In addition, samples of fermented grains were gathered at the early stage of fermentation, and the samples at day 0 and day 2 were gathered on the left side of the X-axis, while the samples at day 21 and day 30 at the end of fermentation were gathered on the right side of the X-axis. The two sets of samples can be separated along the X-axis, indicating a significant difference between the early stage and late stages of fermentation.

### 3.3. VIP Diagram Analysis of OPLS-DA

The VIP map of metabolite OPLS-DA indicates the importance of variables and their contribution to sample differentiation. In Figure 3, the abscess indicates the VIP value, and the ordinate indicates the name of the top 20 differential metabolites. The larger the VIP value, the larger their contribution to the sample differentiation. Generally, variables with default VIP > 1 have significant differences.

The primary metabolites in fermented grains on day 0 and day 2 of the fermentation are shown in Figure 3a, which only the flavin adenine degradation product and xylitol increase continuously from 0 to 2 days of fermentation. The amount of other metabolites decreased continuously on day 2 of fermentation, primarily by a substantial reduction in the amount of organic acids, such as gluconic acid, maleate, oxalate, 4-aminobutyric acid, salicylic acid, threonine, L-malic acid, chlorogenic acid, and allothioneine. There were also sugar derivatives such as acetyl-beta-D-mannosamine and aromatics, such as D-glycerol 1-phosphate, allosine, alpha-lactone, 3-methylcatechol, etc.

The primary metabolites in fermented grains on day 2 and day 10 of the fermentation are shown in Figure 3b. Only 2-deoxyuridine and citric acid were reduced. The largest changes in metabolites remain in organic acids. These include citramalic acid, 3-phenylpyridic acid, maleate, megalomaniac acid, 4-aminobutyric acid, maleic acid, glycine, pimelic acid, 3-hexenylpyridic acid, tartaric acid, and 3-hydroxynorvaline. There was also an increase in several alcohol and glycosidic metabolites, such as palatinose, 2-deoxerythritol, and hydroxyurea-dine. Moreover, melibiose metabolites were also increased from 2 days to 10 days of fermentation.

The primary metabolites in fermented grains on day 10 and day 21 of the fermentation are shown in Figure 3c. However, in addition to the degradation of organic acids from 10 to 21 days of fermentation, there were significant changes in carbohydrate metabolites, such as sucrose, raffinose, galactinol, melibiose, 1-kestose, melezitose, and gentiobiose. The metabolite content gradually decreased from 10 to 21 days of fermentation. Among the acid metabolites, 3-hydroxypropionic acid, 3-phenylpyridic acid, methylmalonic acid, 2-hydroxybutanoic acid, maleic acid, and piperic acid gradually decreased from 10 to 21 days after fermentation. In addition, the metabolite content of cetadiol, trans-3, 5-dimethoxy-4-hydroxycinnamaldehyde, naringin, and hesperidin were all reduced. Only guanine, arachidonic acid, and glutathione gradually increased from 10 to 21 days of fermentation.

The amounts of galactinol, neosperidin, melezitose, salicin, trans-3,5-dimethoxy-4-hydroxycinnamaldehyde, 3-hydroxypropionic aldehyde, sorbitol, methylmalonic acid, 5-dimethoxy-4-hydroxycinnamaldehyde, melibiose, monostearin, cellobiose, gentiobiose, and sophorose increased from 21 to 30 days during fermentation are shown in Figure 3d. The amount of galactose, xylitol, lauric acid, oxalic acid, arachidonic acid, 2-hydroxypyridine, and succinic acid decreased steadily from 21 days to 30 days after fermentation.

In general, sugars and glycols in fermented grains enter the tricarboxylic acid cycle through glycolysis to produce organic acids and other compounds. Some sugar alcohols also have specific health effects, and sorbitol has antioxidant and anti-aging effects [35]. In addition, the succinic acid content at day 30 of fermentation was significantly different. Wang et al. [36] found that succinic acid was one of the leading organic acids in old rice wine through the study of organic acid substances in Macheng old rice wine, consistent with the results of this study. Chun [37] brewed traditional Korean rice wine with different raw materials and studied the physicochemical properties of Sogokju. It has been shown that the organic acids of Sogokju are mainly composed of succinic acid. Wu et al. [38] researched on organic acids and acid-producing bacteria in the fermentation process of Shaoxing rice wine, and found that succinic acid presented a fresh, spicy taste and had a special sour taste in the rice wine, which was mainly generated through the tricarboxylic acid cycle of yeast.

### 3.4. Permutation Test

For the permutation test plot of the experiment, 200 random permutations were performed (Figure 4). The arrows indicated the location of the classification effect for the OPLS-DA model chosen in this experiment. The bar plots showed the distribution of classification effects for the stochastic model of the permutation test. The model performed 200 random permutations and combination experiments on the data, *p* < 0.005. The R2Y and Q2 values were 0.99 and 0.929, respectively, indicating a good model classification.

### 3.5. Heat Map Analysis of Different Compounds in the Fermentation Process of JUNCAO Wine

The relative content of 60 different metabolites was used to plot the heat map for intuitively discovering the changes in differential metabolites during fermentation (Figure 5). The red represents the higher content, the blue represents the lower content, the horizontal axis represents the different fermentation times, and the vertical axis represents the differential metabolites.

The organic acids are one of the chief aromatic substances in liquor. They not only improve the body of the liquor but also have a certain bacteriostasis effect, which has an important effect on the flavor of the liquor [38]. Wu et al. [39] studied the relationship between the content of organic acids and acid-producing bacteria in the fermentation of Shaoxing rice wine. They found that citric acid, oxalic acid, lactic acid, and malic acid increased significantly and extensively in fermentation, among which citric acid and lactic acid increased the fastest in pre-fermentation. Contrary to the results of this study, this may be related to differences in the ingredients and fermentation process. Liu et al. [40] studied the formation mechanism of five unsaturated fatty acids, such as palmitoleic acid and linoleic acid, in the fermentation process of Dong wine. The results showed that the amount reached its maximum on the sixth or seventh day of fermentation and gradually decreased from the eighth day. The rule of variation in unsaturated fatty acids during fermentation is consistent with this study.

In the process of fermentation, sugars and sugar alcohols enter the tricarboxylic acid cycle mainly through glycolysis to further produce organic acids and other compounds [41]. At the same time, it provides energy for microbial growth through the carbohydrate metabolism pathway. Sucrose, galactose, ribose, and xylose were the primary sources of sweetness in JUNCAO wine. 

Amino acids have various flavors, such as sour, sweet, bitter, astringent, and fresh, which can produce higher alcohol through dissimilation, thus enhancing the coordination and consistency of the wine body and contributing to the sensory characteristics of the wine [42]. 

Other differential metabolites detected included 2-hydroxypyridine, thymine, guanine, uridine, salicin, purine riboside, neohesperidin, naringin, pantothenic acid, stigmasterol, loganin, glutathione, trans-3,5-dimethoxy-4-hydroxycinnamaldehyde, and 1-monopalmitin. Glutathione is involved in the tricarboxylic acid cycle and glucose metabolism during biosynthesis. It is a kind of active tripeptide with important physiological functions, mainly composed of glutamic acid, cysteine, and glycine [43]. Yang et al. [44] studied the differences between Feng-flavor base wines of different years using GC-MS technology and found that 1-monopalmitin was not present in new wine. With the extended storage time, glycerol palmitate was first detected in fermented grains, and its formation mechanism needs further investigation.

In addition, we found that at different stages of fermentation, the fermented grain contained a large amount of healthy substances. For example, L-malic acid, which has the aroma and saltiness of apples, is an important intermediate product involved in the tricarbonic acid metabolic pathway and is also one of the body’s essential organic acids, which can improve the utilization of amino acids in the body. In addition, the saltiness is only one-third of that of edible salt. It is often used as an alternative to edible salt in kidney patients and is also used to treat anemia, uremia, high blood pressure, and other conditions. 4-aminobutyric acid can improve nervous function and reduce the occurrence of Alzheimer’s disease, but it also has anti-aging properties, weight loss properties, and improves lipid metabolism. Linolenic acid and arachidonic acid are polyunsaturated fatty acids; linolenic acid is very important for human health, is one of the essential nutrients most people need to supplement, and has the effect of reducing “three high”, anti-inflammatory, anti-aging, and enhanced intelligence and vision; meanwhile, arachidonic acid plays an important role in the prevention of cardiovascular disease, diabetes, and has antitumor properties. Chlorogenic acids are antibacterial, antiviral, antitumor, and free radicals but also have fragrance, color protection and anti-corrosion properties and are widely used in food preservation. Caffeic acid has antiviral and antivenomous effects. Palatinose is a natural sugar found only in trace levels in cane juice and honey. It is widely used in food and drink because it can be fully digested and absorbed by the body while having little effect on the glycemic index. Neohesperidin is a naturally occurring flavonoid compound. Neohesperidin dihydrochalcone, prepared from it, is a functional sweetener with bitterness inhibition and flavor enhancement. Loganin has anti-aging, anti-bacterial, anti-radiation, and other effects and is often used as a cough remedy in traditional Chinese medicine. L-malic acid, 4-aminobutyric acid, and arachidonic acid are common in traditional grain wine. Caffeic acid and chlorogenic acid have been reported in cider and honey mulberry wine [45]. Palatinose, neohesperidin, and loganin have been detected in fermented grains for the first time. As the fermentation time was extended, the amount of palatinose increased. Neohesperidin levels are highest from 2 to 10 days after fermentation, and gradually decrease. The highest levels of loganin were found on day 0 of fermentation, indicating that loganin was initially present in the fermenting raw material and gradually decreased as the fermentation time lengthened. These unique flavor substances may be important features of the flavor of JUNCAO wine, and their role and formation mechanisms in JUNCAO wines need to be further investigated.

### 3.6. Metabolic Pathway Analysis of JUNCAO Wine during Fermentation

The key metabolic pathways were annotated based on the premise that pathway impact values (VIP) > 1 and *p* < 0.05. The metabolic pathways of different metabolites in fermented grains of JUNCAO wine during the fermentation process are shown in Figure 6a. From 0 days to 2 days of fermentation, tricarboxylic acid cycle, metabolism of alanine, aspartic acid and glutamate, β-alanine metabolism, pyrimidine metabolism, and metabolism of glycine, serine, and threonine had a great influence on the fermentation process. As shown in Figure 6b, only three key metabolic pathways, metabolism of alanine, aspartic acid and glutamate, pyrimidine and glycine, and serine and threonine, were present from 2 to 10 days after fermentation. As shown in Figure 6c, the number of key metabolic pathways increased most from 10 to 21 days of fermentation, indicating that microbial metabolism was vigorous during this fermentation. Among them were galactose metabolism, pyruvate metabolism, alanine, aspartic acid and glutamic acid metabolism, pyrimidine metabolism, glycine, serine and threonine metabolism, pentose, glucuronic conversion, oxidative phosphorylation, and steroid biosynthesis. As shown in Figure 6d, pyrimidine metabolism, tricarboxylic acid cycle, alanine, aspartate, and glutamic acid metabolism, pyruvate metabolism, amino and nucleotide sugar metabolism, pentose, glucuronic conversion, and starch and sucrose metabolism increased from 21 to 30 days of fermentation.

These seven metabolic pathways were the critical metabolic pathways at the end of fermentation. These metabolic pathways were similar to the results of previous study [34] on the influence of crucial flavor metabolic characteristics of black glutinous rice wine in different fermentation processes.

The main intracellular metabolic pathways of yeast include the TCA cycle, the glycolysis pathway, the amino acid metabolism pathway, and the fatty acid metabolism pathway. Combined with a comprehensive analysis of differentiated metabolites and key metabolic pathways at different stages of fermentation, it can be seen that polysaccharides are hydrolyzed into disaccharides or tri sugars, and monosaccharides through the tricarboxylic acid cycle, alanine, aspartic acid and glutamic acid metabolism, glycine, metabolism of serine and threonine, galactose, pyruvate, pyrimidine, β-alanine, starch and sucrose, pentose, glucuronic acid conversion and oxidative phosphorylation, and steroid biosynthesis pathway produce energy for microbial growth and multiplication while producing primary metabolites [46,47].

Changes of substances in metabolic pathways were inferred from non-targeted metabolic databases. The metabolic pathway analysis of JUNCAO wine during fermentation is shown in Figure 7. With the extension of fermentation time, the content of sucrose first increased and then decreased in different periods, while the content of sucrose began to decrease rapidly after 10 days of fermentation, and the content of xylose and galactose increased significantly in this period. The reason may be that sucrose, raffinose, and cane triose are produced as galactose by glucose-1-phosphate in starch–sugar metabolism at day 10 under the action of certain enzymes. Except for xylose, which increased to a maximum at the end of fermentation, the levels of melibiose, raffinose, and galactose were shallower at the end of fermentation, which may be because melibiose, raffinose, and galactose generate xylose by the action of galactosidase through the metabolism pathway of galactose. The environment at the end of fermentation may not be suitable for the growth of microorganisms that use xylose, so xylose accumulates in large quantities at the end of fermentation. It was worth noting that galactose levels drop dramatically from 21 to 30 days after fermentation, which may be related to other metabolic pathways. In addition, the ribose content gradually decreased during the 10th day of fermentation and gradually increases after 10 days via the pentose phosphate pathway. As glycolysis or gluconeogenesis progresses in the fermentation process, the polysaccharides were eventually hydrolyzed into monosaccharides to produce pyruvate, which was partially converted into lactate by lactate dehydrogenase. Lactic acid was converted into pyruvate in lactate dehydrogenase and enters the tricarboxylic acid cycle, where the other part of the alanine was converted from pyruvate. Together they entered the tricarboxylic acid cycle to form citric acid, succinic acid, fumaric acid, and L-malic acid. In addition, α-ketoglutaric acid can be metabolized by alanine, aspartic acid, and glutamic acid to produce glutamine, which was further hydrolyzed by glutamine under the action of glutamylase to produce glutamic acid, and then further converted into proline and 4-aminobutyric acid. Proline was not directly involved in yeast metabolism, so it was mainly absorbed by yeast as a nutrient for bacterial growth. Furthermore, 4-aminobutyric acid can be converted to succinic acid with some enzymes. In addition, the contents of L-malic acid and citric acid were the highest at day 0 of fermentation. The contents then gradually decreased as the fermentation time increased, suggesting that they may have come from the raw material. Levels of 4-aminobutyric acid were highest on day 0 of fermentation, decreased significantly on day 2, increased significantly on day 10, and then slowly decreased. However, the relative high content of succinic acid at day 30 at the end of fermentation may be due to the conversion of 4-aminobutyric acid into succinic acid by the metabolic pathway of alanine, aspartic acid, and glutamic acid during fermentation. As a result, the 4-aminobutyric acid content slowly decreased from day 10 of fermentation, while the succinic acid content gradually increased. Fumaric acid, as an unsaturated organic acid, was involved in the fermentation and metabolism of JUNCAO wine and had the aroma of wine. Under the action of certain enzymes, fumaric acid was metabolized by glycine, serine and threonine to produce aspartic acid. Aspartic acid was converted to threonine and uridine by aspartic dehydrogenase, threonine was converted to serine by specific enzymes, and serine and uridine were converted to thymine by several enzymes.

## 4. Conclusions

In this study, based on the non-targeted metabolomics of GC-QTOF-MS, the changes in flavor substances and the enrichment of flavor substance metabolic pathways during the fermentation of JUNCAO wine were revealed for the first time. However, the specific regulatory mechanisms of many metabolites responsible for the flavor of JUNCAO wine are still unknown. For example, palatinose, neohesperidin, and loganin with special effects were discovered for the first time. Subsequent work can provide an in-depth study of research on these specific differential metabolites. The results show that various organic acids, sugars, and sugar-alcohols, amino acids, fatty acids, and aldehydes undergo dynamic changes through metabolic pathways under the action of enzymes at different fermentation stages, and the types and amounts of differential metabolites were not the same. There were 10 vital metabolic pathways in fermentation, including the tricarboxylic acid cycle, alanine, aspartic acid and glutamic acid metabolism, pyrimidine metabolism, and glycine metabolism. The contents of citric acid, L-malic acid, lactic acid, galactose, xylose, 1-kestose, and hydroxyproline changed significantly during the fermentation, which may greatly affect the quality of JUNCAO wine. This paper provides some theoretical guidance for regulating flavor substances in JUNCAO wine.

## Figures and Tables

**Figure 1 foods-12-02254-f001:**
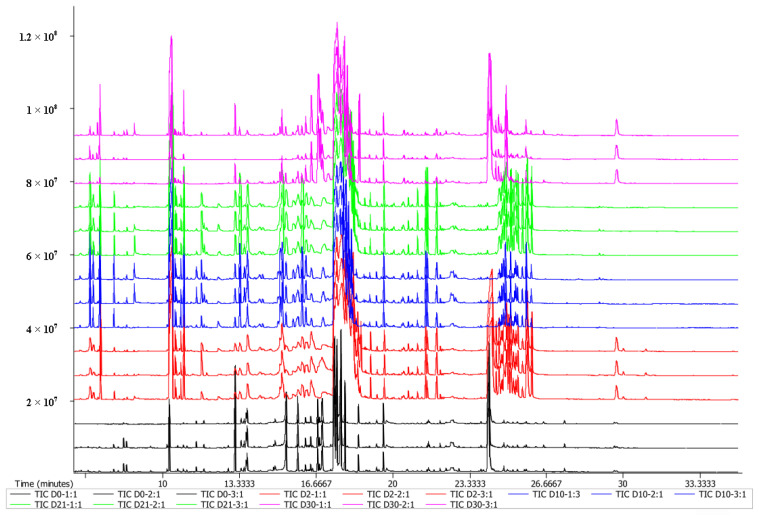
Total ion chromatogram of superimposed QC samples. The ordinate shows the relative intensity abundance, and the abscissa indicates the retention time. (n = 3).

**Figure 2 foods-12-02254-f002:**
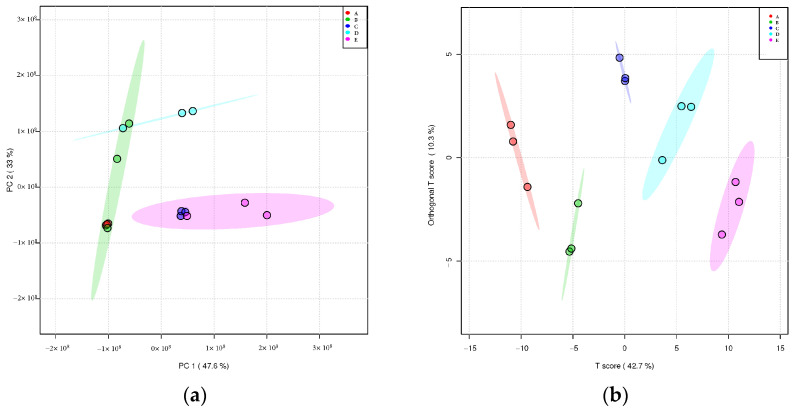
Metabolite PCA map of samples (colored ellipses indicate 95% confidence intervals for metabolites at each stage; A—day 0; B—day 2; C—day 10; D—day 21; E—day 30). (**a**) PCA score of metabolites in fermented grains during fermentation and (**b**) OPLS-DA score plot of metabolites of fermented grains during the fermentation process.

**Figure 3 foods-12-02254-f003:**
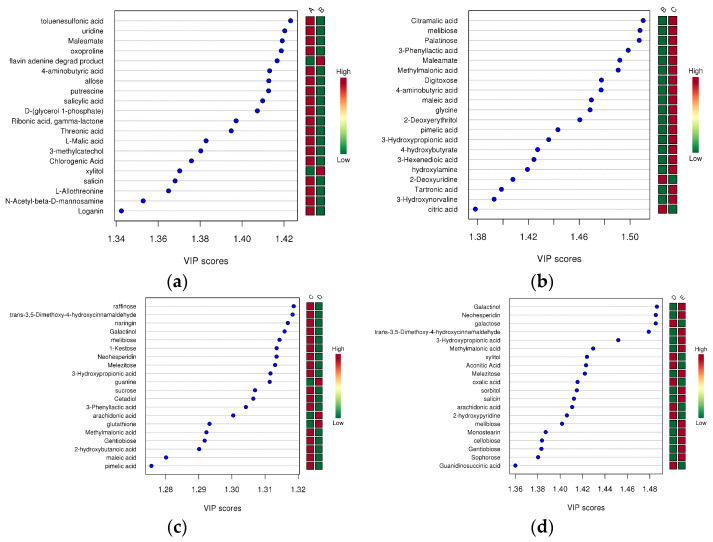
VIP plot analysis of metabolite OPLS-DA (A—day 0; B—day 2; C—day 10; D—day 21; E—day 30). (**a**) OPLS−DA VIP map of metabolites in fermented grains from day 0 to day 2 of fermentation, (**b**) OPLS−DA VIP map of metabolites in fermented grains from day 2 to day 10 of fermentation, (**c**) OPLS−DA VIP map of metabolites in fermented grains from day 10 to day 21 of fermentation, and (**d**) OPLS−DA VIP map of metabolites in fermented grains from day 21 to day 30 of fermentation.

**Figure 4 foods-12-02254-f004:**
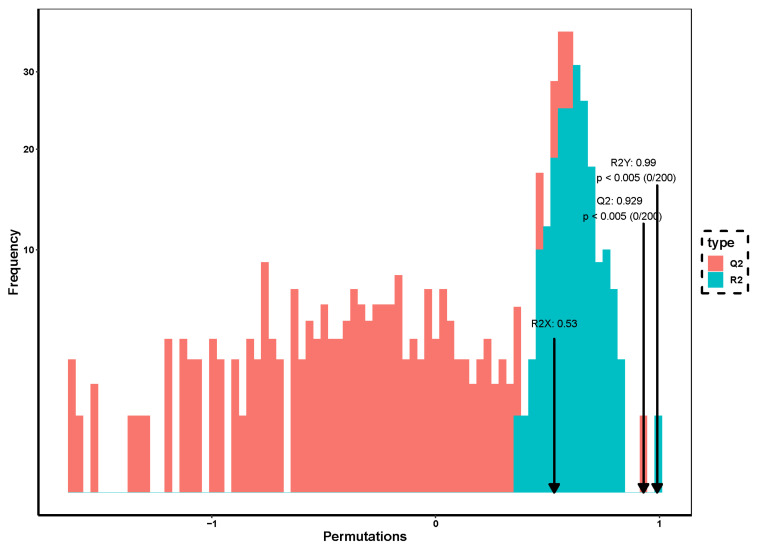
Permutation test of the OPLS−DA model.

**Figure 5 foods-12-02254-f005:**
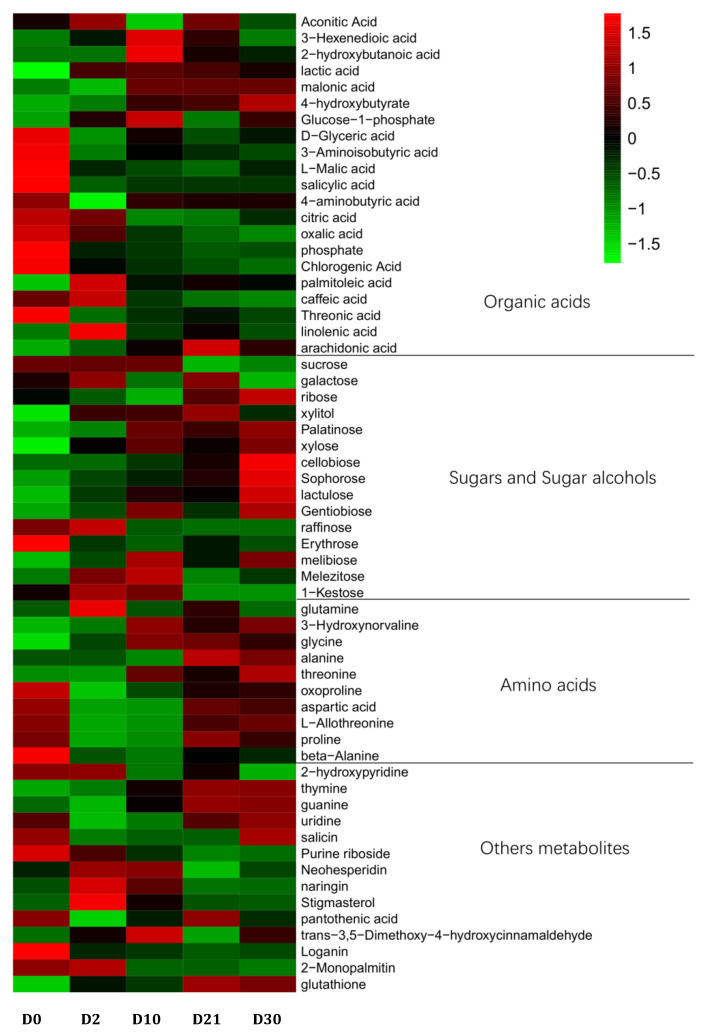
The relative contents of different metabolites at different fermentation stages.

**Figure 6 foods-12-02254-f006:**
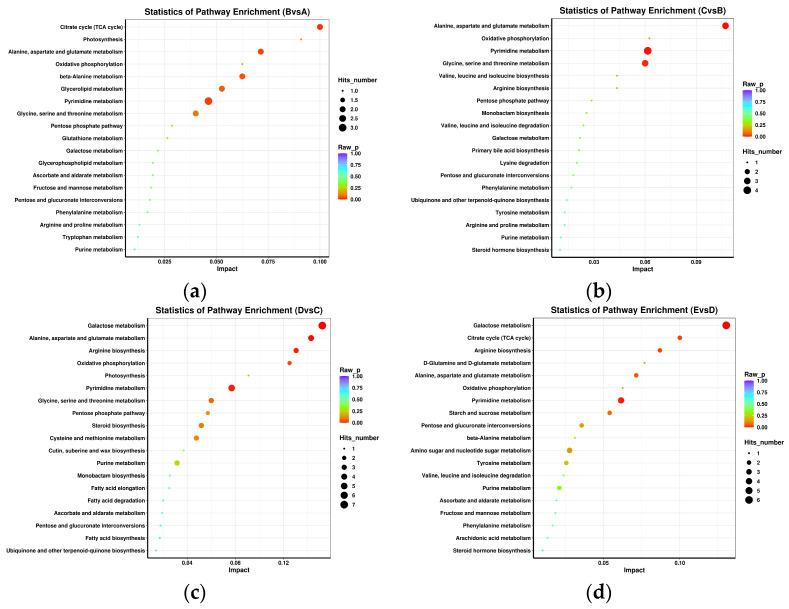
KEGG enrichment analysis of metabolites with differential accumulation. (A—day 0; B—day 2; C—day 10; D—day 21; E—day 30.) (**a**) Enriched bubble diagram of differential metabolite pathways from 0 days to 2 days, (**b**) enriched bubble diagram of differential metabolite pathways from 2 days to 10 days, (**c**) enriched bubble diagram of differential metabolite pathways from 10 days to 21 days, and (**d**) enriched bubble diagram of differential metabolite pathways from 21 days to 30 days.

**Figure 7 foods-12-02254-f007:**
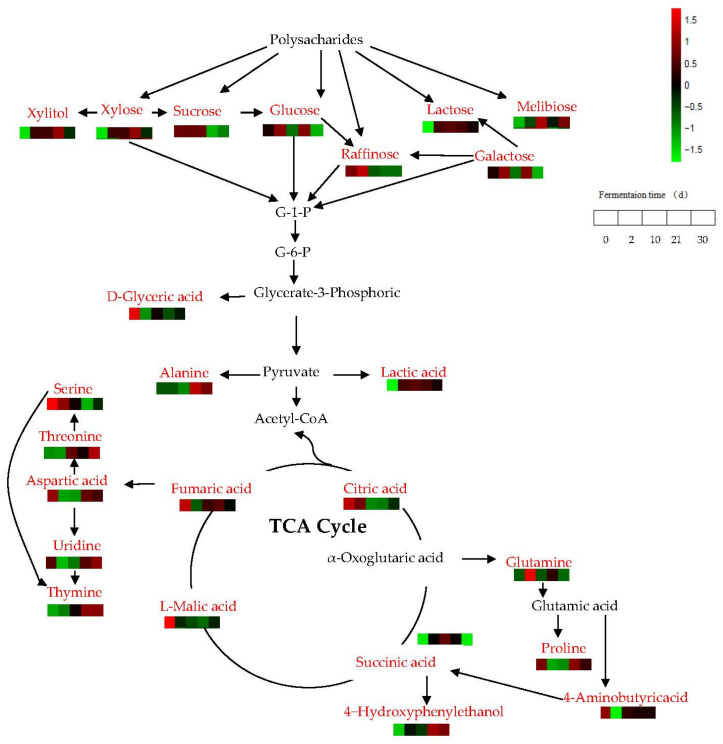
Analysis of metabolic pathways in the fermentation process of JUNCAO wine. The red names represent metabolites that changed obviously in different fermentation stages of JUNCAO wine, while the black names represent compounds that had no significant difference.

## Data Availability

The data presented in this study are available on request from the corresponding author.

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
