# Peer review of "Analysis of Metabolic Components of JUNCAO Wine Based on GC-QTOF-MS"

_foods, 2023, doi:10.3390/foods12112254_

Round 1

Reviewer 1 Report

The manuscript submitted by Fan et al reads good and interesting. Below are my comments/suggestions to improve the quality of the manuscript.

Abstract - write the full form of PCA, VIP

Text - line 41, 45, 91, 95 - no need to write the full scientific name ex: C. fungigraminus and C. citratus

Line 47 - what does mean by JUNCO aroma? 

Is there any reports regarding the microbes that responsible for JUNCO fermentation?

line 75 - remove "gas"

Line 92 - should be corrected as National Engineering Research Center

Line 103 - full form of EP tube

Line 114 - what are the components of the QC samples?

Section 2.3 - MS parameters? add after GC parameters.

Line 131 - how does the peak extraction, baseline correction… were carried out? add details to SI file.

Please plot and include the chromatograms of the QC samples through out the experiments into the SI file

Line 211 and 213 - D0 and D2 means days 0 and 2 I guess. Use same format to understand clearly.

I think authors did not run the standard compounds to specifically identify the preliminary identified compounds.. Thus we can not used the term "IDENTIFIED METABOLITES' it should be ANNOTATED METABOLITES' Please change the wordings. 

Also, I suggest strongly to go through the 10.1007/s11306-007-0082-2 Metabolomics Standard Initiative and report the data and validate the methods accordingly.

Minor English editing and reviewing is required. 

Author Response

Dear Editor and Reviewers:
Thanks for your letter and comments concerning our manuscript entitled “Analysis of metabolic components of JUNCAO wine based on GC QTOF MS” (No.: foods 2354671). Those comments are all valuable and very helpful for revising and improving our paper. We have studied the comments carefully and made correction according to these comments . The main corrections in the paper and the responds to the c omments are as follow s [Referee comments 1
1. Abstract write the full form of PCA, VIP
Reply:
Thanks for your reminder. The full form s of PCA and VIP have been added in the abstract.
2. Text line 41, 45, 91, 95 no need to write the full scientific name ex: C. fungigraminus and C. citratus
Reply:
Thanks for your reminder. All of Cenchrus fungigraminus Z. X. Lin & D. M. Lin & S. R. Lan sp. Nov. and Cymbopogon citratus Stapf have been changed to C. fungigraminus and C.citratus Line 41,4 5, 91,94 in the ne w manuscript.
3. Line 47 what does mean by JUNCO aroma?
Is there any reports regarding the microbes that responsible for JUNCO fermentation?
Reply:
JUNCO aroma i s mean a strong and distinctive grass aroma as described in r eference 7 We have also tested the fermentation microbiota of JUNCO wine, and the paper containing relevant data is still under review. If you feel it is necessary, after the publication of the paper containing microbial community informa tion, we will add the literature as a reference to the subsequent paper revisions.
4. line 75 remove "gas"
Reply:
Thanks for your reminder. The "gas" has been remove d. Lin e 74 in the ne w manuscript.
5. Line 92 should be corrected as National Engineering Research Center
Reply:
Thanks for your reminder. It has been corrected as “National Engineering Research
Center” Line 92 in t he new manuscript.
6. Line 103 full form of EP tube
Reply:
Thanks for your reminder. We have added the full form of EP tube as E p pendorf EP ) tube
L ine 106 in th e new manuscript.
7. Line 114 what are the components of the QC samples?
Reply:
QC samples is an equal volume mixture of all scientific samples to be tested. In this study, each sample w as taken 20 μ L mix for QC. QC samples are used for testing before sample status and balance chromatography mass spectrometry system, and to evaluate the system stability, in the experimental process
8. Section 2.3 Section 2.3 -- MS parameters? MS parameters? AAdd after GC parameters.dd after GC parameters.
Reply:
Reply: Thanks for your reminThanks for your reminder. der. We added the We added the MS parametersMS parameters afterafter GC parameteGC parametersrs (line 131(line 131--134 134 in the new manuscriptin the new manuscript) as ) as ““The injection, transfer line, and ion source temperatures were 280, 280, The injection, transfer line, and ion source temperatures were 280, 280, and 250°C, respectively. The energy was and 250°C, respectively. The energy was --70eV in electron impact mode. The mass spectromet70eV in electron impact mode. The mass spectrometry ry data were acquired in fulldata were acquired in full--scan mode with the m/z range of 50scan mode with the m/z range of 50--500 at a rate of 20 spectra per 500 at a rate of 20 spectra per second after a solvent delay of 6.17min.second after a solvent delay of 6.17min.””
9. Line 131 Line 131 -- how does the peak extraction, baseline correction… were carried out?how does the peak extraction, baseline correction… were carried out?
Reply:
Reply: We carried out We carried out peak peak extraction, baseline correction, peak integration, peak alignment, and extraction, baseline correction, peak integration, peak alignment, and other statistical analysisother statistical analysis according to theaccording to the rreference “FiehnLib: mass spectral and retentioneference “FiehnLib: mass spectral and retention index index libraries for metabolomics based on quadrupole and timelibraries for metabolomics based on quadrupole and time--ofof--flight gasflight gas chromatography/mchromatography/mass ass spectrometryspectrometry byby Kind et al.Kind et al., in , in AnalAnalyticalytical ChemChemistry,istry, 20092009,, 8181,, 1003810038--48.”48.” We added this We added this referencereference in thein the new manuscript. new manuscript. Line Line 139139--140140,, referencereference 2828..
10. Please plot and include the chromatograms of the QC samples through out the Please plot and include the chromatograms of the QC samples through out the experiments into the SI fexperiments into the SI fileile..
Reply:
Reply: Thanks for your reminder.Thanks for your reminder. The The cchromatograms of QC samples throughout the experimenthromatograms of QC samples throughout the experimentss have beenhave been plotplottted and included in the SI fileed and included in the SI file ((Appendix BAppendix B).).
11. Line 211 and 213 Line 211 and 213 -- D0 and D2 means days 0 and 2 I guess. Use same format to D0 and D2 means days 0 and 2 I guess. Use same format to understand clearly.understand clearly.
Re
Reply:ply: Thanks for your reminder. Thanks for your reminder. D0D0, , D2D2, , DD10,10, and D2and D21 have been replaced as 1 have been replaced as day 0day 0,, day day 22, day , day 10 and day 2010 and day 20. . LinLinee 220, 222220, 222 in tin the new manuscript.he new manuscript.
12. I think authors did not run the standard compounds to specifically identify the I think authors did not run the standard compounds to specifically identify the preliminary identified compounds. Thus we can not used the term "IDENTIFIED preliminary identified compounds. Thus we can not used the term "IDENTIFIED METABOLITES' it should be ANNOTATED METABOLITES' Please change the METABOLITES' it should be ANNOTATED METABOLITES' Please change the wordings. wordings.
Reply:
Reply: Thanks for yourThanks for your reminder. reminder. The The ““identified metabolitesidentified metabolites”” have been have been corrected ascorrected as ““annotated annotated metabolitesmetabolites””.. LinLine e 20, 23, 36320, 23, 363 in the new main the new manuscript.nuscript.
13. Also, I suggest strongly to go through the 10.1007/s11306Also, I suggest strongly to go through the 10.1007/s11306--007007--00820082--2 Metabolomics 2 Metabolomics Standard Initiative and report the data and Standard Initiative and report the data and validate the methods accordingly.validate the methods accordingly.
Reply: Thanks for your
Reply: Thanks for your valuable commentsvaluable comments. . We have carefully studied the literature you We have carefully studied the literature you recommended, which outlines the general analytical process of metabolomics. Our metabolomics recommended, which outlines the general analytical process of metabolomics. Our metabolomics analysis method also referanalysis method also referrreedd to to aa large number of metabolomics literaturelarge number of metabolomics literaturess and is currently a and is currently a popular analysis method.popular analysis method. The literatureThe literaturess similar to our analysis method similar to our analysis method areare as follows:as follows: [1] [1] Chong, Chong, J; Yamamoto, M; Xia, JG. MetaboAnalystR 2.0: From Raw Spectra to Biological InsightJ; Yamamoto, M; Xia, JG. MetaboAnalystR 2.0: From Raw Spectra to Biological Insights. s. MetaboliteMetabolites, 2019; 9 (3)s, 2019; 9 (3); [2] ; [2] Garcia A, Barbas C. Gas chromatographyGarcia A, Barbas C. Gas chromatography--mass spectrometry mass spectrometry (GC(GC--MS)MS)--based metabolomics. Methods Mol Biol, 2011, 708(191based metabolomics. Methods Mol Biol, 2011, 708(191--204)204); [3] ; [3] Saccenti E, Hoefsloot Saccenti E, Hoefsloot H C, Smilde A K, et al. Reflections on univariate and multivariate analysis of metaboloH C, Smilde A K, et al. Reflections on univariate and multivariate analysis of metabolomics data. mics data. MeMetabolomics, 2014, 10(3): 361tabolomics, 2014, 10(3): 361--7474; [4] ; [4] Aviram R, Manella, G, Kopelman, NM, et al. Lipidomics Aviram R, Manella, G, Kopelman, NM, et al. Lipidomics Analyses Reveal Temporal and Spatial Lipid Organization and Uncover Daily Oscillations in Analyses Reveal Temporal and Spatial Lipid Organization and Uncover Daily Oscillations in
Intracellular Organelles[J]. Molecular Cell, 2016, 62, 636
Intracellular Organelles[J]. Molecular Cell, 2016, 62, 636--648648; [5] ; [5] KindKind,, TT.;.; Wohlgemuth GWohlgemuth G.;.; Lee Lee DD..YY.; .; Lu, YLu, Y..; ; Palazoglu, MPalazoglu, M.; .; Shahbaz, SShahbaz, S.; .; Fiehn, OliverFiehn, Oliver. FiehnLib: mass spectral and retention. FiehnLib: mass spectral and retention index libraries for metabolomics based on quadrupole and timeindex libraries for metabolomics based on quadrupole and time--ofof--flight gasflight gas chromatography/chromatography/ mass spectrometry. Analmass spectrometry. Anal.. ChemChem.. 20020099, , 8181, , 1003810038--4848.. Nevertheless, we still believe that your Nevertheless, we still believe that your suggestion is very valuable. But due to time constraints, we have not had time to conduct suggestion is very valuable. But due to time constraints, we have not had time to conduct the the additional analysisadditional analysis until nowuntil now. If you . If you still still think it is necessary, we will add think it is necessary, we will add the the relevant analysis relevant analysis processprocess you you recommended in the subsequent paper revisions.recommended in the subsequent paper revisions.

Reviewer 2 Report

Overview

Fan et al. submitted the article “Analysis of metabolic components of JUNCAO wine based on  

GC-QTOF-MS” where they presented the dynamic metabolite changes during the fermentation of JUNCAO wine.

There is a good overall description about the aims and the data analysis done to better highlight all the metabolic differences during the fermentation JUNCAO wine.

I do think the paper is too long especially the heat map discussion and the metabolic pathway. I would suggest reducing these parts. For example, all the discusses parts about sugar can be moved in the SI or appendix so you can focus only on the differences you observed.

Comments:

Line 56: It would be good to mention also the untargeted high resolution mass spectrometry techniques as for untargeted metabolomics as the one presented by Onzo et al. in Analytical and Bioanalytical Chemistry414(27), 7805-7812. On the analysis of wine samples.

Line 86: Please specify BSTFA 86 (containing 1% TMCS, v/v)

Line 109: Please give more info regarding the derivatization process.

Line 132: explain GDM database.

Line 133: please check the sentences.

Line 140: provide more information about the fermentation process (add a sentence about it)

Line 147-152: I don’t think this part is necessary.

Figure 1: please revise the caption. Graph shows the relative intensity abundance.

Line 186- 189: it would be good to report these data.

Paragraph 3.3: please check punctuation, as some sentences are not clear.

Line 241: check trans-3.

Line 436: it should be 3.6, please correct.

Figure 7: please explain why you use black or red for the compound’s names.

English quality is on the average.

Author Response

Thanks for your letter and comments concerning our manuscript entitled “Analysis of metabolic components of JUNCAO wine based on GC-QTOF-MS” (No.: foods-2354671). Those comments are all valuable and very helpful for revising and improving our paper. We have studied the comments carefully and made correction according to these comments. The main corrections in the paper and the responds to the comments are as follows:

[Referee comments 2]

  1. I do think the paper is too long especially the heat map discussion and the metabolic pathway. I would suggest reducing these parts. For example, all the discusses parts about sugar can be moved in the SI or appendix so you can focus only on the differences you observed.

Reply: Thanks for your valuable comments. We have streamlined the heat map discussion and metabolic pathway. The detailed discussion about organic acids, amino acids, sugars, sugar alcohols, and other compounds have been moved to the Appendix C.

  1. Line 56: It would be good to mention also the untargeted high resolution mass spectrometry techniques as for untargeted metabolomics as the one presented by Onzo et al. in Analytical and Bioanalytical Chemistry, 414(27), 7805-7812. On the analysis of wine samples.

Reply: Thank you very much for the recommendation. The suggested reference is an important untargeted metabolomic analysis method for wine samples, and we added it in line 57 in the new manuscript (reference 11).

  1. Line 86: Please specify BSTFA (containing 1% TMCS, v/v).

Reply: “BSTFA (containing 1% TMCS, v/v)” have been specified as “bis(trimethylsilyl)trifluoroacetamide (BSTFA) and trimethylchlorosilane (TMCS) (BSTFA/TMCS: 99/1, v/v)”. Line 114-115 in the new manuscript.

  1. Line 109: Please give more info regarding the derivatization process.

Reply: The derivatization process for details is added in paragraph 2.3.

  1. Line 132: explain GDM database.

Reply: Thanks for your reminder. The Golm Metabolome Database (GMD) is a metabolomics database established by scientists at the Max Planck Institute in Germany, containing 1450 identified metabolites and 10336 related mass spectra. The resources focus on untargeted metabolomics of GC-MS, with the greatest feature being the GC-MS spectrum containing a large amount of plant metabolites (especially after derivatization). Users can import their own GC-MS data for search, comparison, and identification. We have added its full name and reference to the new manuscript (line 140-142, reference 29).

  1. Line 133: please check the sentences.

Reply: Thanks for the reminder. The sentence has been corrected as: The ratio of peak area of target compounds to internal standard was analyzed in Microsoft Excel to obtain the relative content of metabolites. (Line in the new manuscript)

  1. Line 140: provide more information about the fermentation process (add a sentence about it)

Reply: Thanks for your reminder. We have added the detailed information about the fermentation process to the new manuscript, line 95-99.

  1. Line 147-152: I don’t think this part is necessary.

Reply: This part is an explanation of the horizontal and vertical coordinates of Figure 1. It has been deleted in the new manuscript.

  1. Figure 1: please revise the caption. Graph shows the relative intensity abundance.

Reply: The title of Figure 1 has been revised as “Total ion chromatogram of superimposed QC samples. The ordinate shows the relative intensity abundance, and the abscissa indicates the retention time.”

  1. Line 186- 189: it would be good to report these data.

Reply: Thanks for your reminder. We have added “The differences between the groups can be seen in the abscissa direction (42.7%), and the differences in samples within the group can be seen on the vertical axis (10.3%).” into the new manuscript, line 197-199.

  1. Paragraph 3.3: please check punctuation, as some sentences are not clear.

Reply: Thanks for your reminder. The punctuation in the paragraph 3.3 has been checked.

  1. Line 241: check trans-3.

Reply: Thanks for your reminder. “trans-3” has been corrected as “trans-3, 5-dimethoxy-4 hydroxycinnamaldehyde”, line 243-244 in the new manuscript.

  1. Line 436: it should be 3.6, please correct.

Reply: Thanks for your reminder. “3.5” have been corrected as “3.6”, line 364 in the new manuscript.

  1. Figure 7: please explain why you use black or red for the compound’s names.

Reply: The red names represent metabolites that changed obviously in different fermentation stages of JUNCAO wine, while the black names represent compounds that had no significant difference. Line 475-477 in the new manuscript.

Round 2

Reviewer 1 Report

Dear Authors, 

thank you for revising the manuscript according to our suggestions. I prefer to see revised manuscript with and without track changes for better clarity. But I understand that Authors has gone through every detail and revised accordingly.

For my comment on following the Metabolomoics Stnd Initiative , I still feel that validation of annotated compound is a must. Thus I wanted to request the Authors to group the annotated metabolites according to Metabolite Identification Levels.  

Minor English language editing is required.  

Author Response

   Thank you for your letter and for your comments concerning our manuscript entitled “Analysis of metabolic components of JUNCAO wine based on GC-QTOF-MS” (No.: foods-2354671). Those comments are valuable and very helpful for revising and improving our paper, as well as the important guiding significance to our researches. We have studied comments carefully and have made correction which we hope to meet with approval. The main corrections in the paper and the responds to the comments are as following:

[Referee comments]

  1. For my comment on following the Metabolomoics Stnd Initiative, I still feel that validation of annotated compound is a must. Thus I wanted to request the Authors to group the annotated metabolites according to Metabolite Identification Levels.

Reply: Thanks for your valuable comments. We appreciate your suggestion regarding the validation of annotated compounds and their grouping based on Metabolite Identification Levels. This will facilitate the streamlining of the quality assurance process and enhance the ability to discern confirmation levels for each metabolite, while also establishing a shared language among metabolomics researchers for communication and comparison of results. Furthermore, it will enable the creation of a comprehensive database of validated metabolites that can serve as a valuable resource for future studies in this field. Ultimately, grouping annotated metabolites according to Metabolite Identification offers numerous benefits.

   This study the fermented grains were shipped to the Beijing Allwegene Technology Company, Beijing, China for analysis. In the substance identification phase, due to the absence of standardized protocols, reliance is placed solely on metabolite information in databases. Therefore, ensuring database accuracy is a crucial consideration for identifying non-targeted metabolites. In this research the volatile compounds were identified using comparison with mass spectra in the National Institute of Standards and Technology library (NIST). During the data acquisition stage, both parent ions and fragment ions are obtained via a non-targeted metabolic data dependent acquisition (DDA) model. Unlike the targeted MRM mode, selection of parent ions for fragmentation and secondary scanning is based on their signal strength. According to the set threshold for parent ion signal, those meeting the range are subjected to secondary fragmentation, resulting in a higher number of detected substances compared to targeted metabolites.

   Proposed minimum reporting standards for chemical analysis as the one presented by Lloyd et al. in Metabolomoics, 3:211-221. This article proposes the minimum reporting standards related to the chemical analysis aspects of metabolomics experiments including: sample preparation, experimental analysis, quality control, metabolite identification, and data pre-processing. The article mentions that if spectral (MS or NMR) matching is utilized in the identification process then the authentic spectra used for the spectral matching should be described appropriately or libraries made publicly available. It is preferred that the reference spectra are made available at no cost, but the Chemical Analysis Working Group (CAWG) recognizes that this may not always be possible for commercialized libraries (NIST, Wiley, etc.). If choose not to provide the experimental evidence to support the identifications, then the identifications should be reported as ‘putative identifications’. Therefore, we will make the necessary changes to the terminology in the revised manuscript. Instead of using "identified metabolites," we will use the term "annotated metabolites" to accurately reflect the fact that the identity of these compounds has not been specifically confirmed. Some phrases in the article are euphemistic, such as the final analysis of metabolic pathways, which refers to "changes in substances involved in metabolic pathways inferred from non-targeted metabolic databases." Line 106, 107, 144, 145, 407 and 408 in the new manuscript. In addition, the possible differential metabolites during the fermentation of JUNCAO wine included in the SI file (Appendix D).

   Your input is highly valued, and we will take it into consideration as we work towards addressing this matter in our future revisions.
